# Association between Timing of Epinephrine Administration and Outcomes of Traumatic Out-of-Hospital Cardiac Arrest following Traffic Collisions

**DOI:** 10.3390/jcm11123564

**Published:** 2022-06-20

**Authors:** Sanae Hosomi, Tetsuhisa Kitamura, Tomotaka Sobue, Ling Zha, Kosuke Kiyohara, Tasuku Matsuyama, Jun Oda

**Affiliations:** 1Department of Traumatology and Acute Critical Medicine, Osaka University Graduate School of Medicine, 2-15 Yamada-oka, Suita 565-0871, Japan; odajun@gmail.com; 2Division of Environmental Medicine and Population Sciences, Department of Social Medicine, Osaka University Graduate School of Medicine, 2-2 Yamadaoka, Suita 565-0871, Japan; lucky_unatan@yahoo.co.jp (T.K.); tsobue@envi.med.osaka-u.ac.jp (T.S.); ivy_mist@outlook.com (L.Z.); 3Department of Food Science, Faculty of Home Economics, Otsuma Women’s University, 12 Sanban-cho, Chiyoda-ku, Tokyo 102-8357, Japan; kiyosuke0817@hotmail.com; 4Department of Emergency Medicine, Kyoto Prefectural University of Medicine, Kamigyo-ku, Kyoto 602-8566, Japan; task-m@koto.kpu-m.ac.jp

**Keywords:** traffic collision, mortality, out-of-hospital cardiac arrest, trauma, Japan

## Abstract

The effects of epinephrine administration timing on patients with out-of-hospital cardiac arrest (OHCA) following traffic collisions are unknown. We analyzed the 2013–2019 All-Japan Utstein Registry data of 2024 such patients aged ≥18 years who were resuscitated by emergency medical service (EMS) personnel or bystanders and then transported to medical institutions. Time from 119 call to epinephrine administration was classified into quartiles: Q1 (6–21 min), Q2 (22–26 min), Q3 (27–34 min), and Q4 (35–60 min). Multivariable logistic regression analysis was used to assess the effects of epinephrine administration timing on one-month survival after OHCA. Overall, the one-month survival rates were 3.2% (15/466) in Q1, 1.1% (5/472) in Q2, 1.9% (11/577) in Q3, and 0.2% (1/509) in Q4. Additionally, the one-month survival rate decreased significantly in the Q4 group (adjusted odds ratio, 0.07; 95% confidence interval, 0.01–0.57) compared with the Q1 group, and the probability of one-month survival decreased as the time from the EMS call to epinephrine administration increased (*p*-value for trend = 0.009). Only four patients (0.9% [4/466]) with the earliest epinephrine administration showed a good neurological outcome.

## 1. Introduction

Traffic collisions are a major cause of hospitalizations and deaths worldwide, resulting in a huge socioeconomic burden [1,2]. The Global Status Report on Road Safety 2018 reported 1.35 million annual road accident-related deaths [3]. The burden is disproportionately borne by developing countries, which have higher rates. However, this also remains an issue in developed countries and the estimated mortality rates in Europe and the United States are 11.7 and 17 per 100,000 persons, respectively [3]. Despite recent advances in driver monitoring and safety assistance control systems, the survival rates after out-of-hospital cardiac arrest (OHCA) related to road accidents remain low, even with maximal resuscitative efforts [4,5].

Although epinephrine is one of the most widely used resuscitation drugs worldwide, the current literature suggests that epinephrine may only provide benefits in certain situations [6]. Its benefits and risks remain controversial in the context of trauma [7,8]. A large randomized controlled trial assessing survival from OHCA after pre-hospital drug administration including epinephrine failed to show decreased mortality rates and concluded that epinephrine should not be recommended in a pre-hospital setting [9]. However, previous studies have shown that earlier epinephrine administration is associated with improved survival from OHCA with both shockable and unshockable initial rhythms [10,11].

Furthermore, the effectiveness of epinephrine timing on patients with traumatic OHCA following traffic collisions has not been determined in previous studies [12,13]. Given that the effectiveness of epinephrine is time-dependent, early administration may improve survival in medical institutions. This study aimed to evaluate the association between the timing of the first dose of epinephrine and the outcomes of traumatic OHCA following traffic collisions.

## 2. Materials and Methods

### 2.1. Study Design and Setting

The All-Japan Utstein Registry is a prospective, population-based registry of OHCA based on the standardized Utstein Style [14,15]. However, the evaluation of the detailed causes of noncardiac OHCA did not begin until 2013. The study enrolled adults aged ≥18 years with OHCA after traffic collisions and before the arrival of emergency medical services (EMS), who were resuscitated by bystanders or EMS personnel and transported to medical institutions in Japan between 1 January 2013, and 31 December 2019. In this study, we excluded pediatric cases, as the outcomes and characteristics of OHCA differ between children and adults [16,17].

Cardiac arrest was defined as the cessation of cardiac mechanical activity, as confirmed by the absence of signs of circulation [18,19]. In the registry, cardiac arrests were classified based on either cardiac or noncardiac origins, with the latter including those resulting from asphyxia, cerebrovascular disease, external causes, malignant tumors, accidental hypothermia, anaphylaxis, drug overuse, traffic collisions, and other causes. Diagnoses were made clinically by the physician treating the patient who collaborated with the EMS personnel. In this study, only patients with traumatic OHCA following a traffic collision were included. The exclusion criteria were as follows: no resuscitation, unknown bystander cardiopulmonary resuscitation (CPR), unknown first cardiac rhythm, time from the 119 call to epinephrine administration >60 min, and no epinephrine use.

### 2.2. EMS Organization in Japan

Details of the EMS system in Japan have been described previously [18,19]. In brief, the EMS system is operated by local fire stations, and when called, an ambulance is dispatched from the nearest fire station. The EMS provides emergency services 24 h a day. Most highly trained pre-hospital emergency care providers are called emergency life-saving technicians (ELSTs). Usually, each ambulance carries a crew of three emergency providers including at least one ELST [20]. ELSTs are allowed to insert intravenous lines and adjunct airway devices as well as the use of semi-automated external defibrillators for patients with OHCA. In Japan, EMS personnel are not allowed to perform advanced interventions such as drug administration (except for adrenaline) and needle decompression for tension pneumothorax. Additionally, specially trained ELSTs have been permitted to perform tracheal intubations since July 2004 and administer intravenous epinephrine since April 2006. Patients aged ≥18 years received ≥1 dose of adrenaline (1 mg bolus) during resuscitation. Living wills or do-not-resuscitate orders are not generally accepted in Japan, and EMS providers are not enabled to terminate resuscitation at the scene. Therefore, almost all patients with OHCA treated by EMS teams were transported to a hospital and enrolled in the All-Japan Utstein Registry, excluding those with incineration, decapitation, decomposition, dependent cyanosis, or rigor mortis.

The use of an automated external defibrillator (AED) by citizens has been legally accepted since July 2004. In Japan, approximately two million citizens participate in community CPR programs yearly, which include training in mouth-to-mouth ventilation, chest compressions, and AED use [17,18,19,20]. All EMS providers perform CPR according to the Japanese CPR guidelines [21].

### 2.3. Data Collection and Quality Control

Data were collected prospectively using a form that included information recommended in the Utstein Style reporting guidelines for cardiac arrest [14,15]. Age, sex, first recorded cardiac rhythm, type of bystander witness status, life support by EMS personnel (i.e., use of insertion of an intravenous line, advanced life support [ALS] devices), the time course of epinephrine administration and resuscitation, pre-hospital return of spontaneous circulation (ROSC), and one-month survival were obtained. Additionally, the times of 119 calls, ambulance arrival at the site of the incident, contact with patients, CPR initiation, defibrillation delivered by EMS personnel, and arrival at the hospital were recorded using the clock of each EMS system. Essentially, the first recorded rhythm was judged by the EMS monitor. While using a public access AED, the EMS teams recorded rhythm such as ventricular fibrillation (VF)/pulseless ventricular tachycardia (pVT), pulseless electrical activity (PEA), and asystole using the monitor after they arrived on the scene. In cases of shock performed by bystanders using a public access AED, the patient’s first recorded rhythm was regarded as pVT or VF. The type of bystander CPR was obtained through observation and interviews with bystanders conducted by the EMS personnel before leaving the accident scene. The forms were completed by EMS personnel in cooperation with the treating physicians in charge, and the data were integrated into the All-Japan Utstein Registry database server and logically checked by the computer system. When data forms were incomplete, the Fire and Disaster Management Agency requested the provision of missing data from the respective fire stations.

All survivors of OHCA were followed up for up to one month after cardiac arrest by the EMS personnel in charge. The one-month neurological outcome was determined by the treating physician using the cerebral performance category (CPC) scale, which included the following categories: Category 1, good cerebral performance; Category 2, moderate cerebral disability; Category 3, severe cerebral disability; Category 4, coma or vegetative state; and Category 5, death [14,15].

### 2.4. Outcome Measures

The main outcome measure was one-month survival. The secondary outcomes included pre-hospital ROSC and one-month survival with favorable neurological outcomes, which were defined as CPC Categories 1 or 2 [14,15].

### 2.5. Statistical Analysis

Categorical variables are presented as counts with proportions, and the χ^2^ test was used to compare the four groups. Continuous variables are presented as medians with interquartile ranges, and the Kruskal–Wallis test was used to compare the four groups.

In addition to the minute-by-minute classification, we tetrachotomized time from the 119 call to epinephrine administration on the basis of patients who received the initial epinephrine. To examine the relationship between outcomes and epinephrine timing, we evaluated epinephrine timing as a categorical variable in multivariable logistic regression models, and the adjusted odds ratios (ORs) and their 95% confidence intervals (CIs) were calculated. Factors that were associated with clinical outcomes and biologically essential were included as potential confounders in multivariable analyses [4,5,6,7,8,9]. These variables included age (18–64, 65–74, ≥75 years), sex (male, female), witness status (none, witnessed by bystanders), first documented rhythm (VF/pVT, PEA, asystole), bystander CPR status (no CPR, any CPR), advanced airway management, (none, endotracheal intubation [ETI], supraglottic airway [SGA]), pre-hospital physician involvement, (i.e., bias from another ALS procedure such as insertion of chest tube, blood transfusion, resuscitative endovascular occlusion of the aorta, and thoracotomy), and year of the arrest. We also evaluated epinephrine timing as a continuous variable in the multivariable logistic regression models. In the subgroup analysis, we conducted a multivariate analysis of one-month survival from OHCA after dividing the patients into three age groups: 18–64, 65–74, and ≥75 years.

All statistical analyses were conducted using STATA (version 17; StataCorp LP, College Station, TX, USA). All tests were two-tailed, and *p*-values of <0.05 were considered as statistically significant.

### 2.6. Ethics Approval

This manuscript was written based on the STROBE statement for the reporting of cohort and cross sectional studies [22]. The study design was approved by the Ethics Committee of the Osaka University Graduate School of Medicine (approval number: 14147). The requirement for written informed consent was waived because of the retrospective nature of the study. The All-Japan Utstein Registry records did not include personal identifiers.

## 3. Results

### 3.1. Eligible Patients

An overview of the patients based on the Utstein template is shown in Figure 1. The data of 868,065 adults with cardiac arrests were documented during the seven year period. Resuscitation was attempted in 847,416 patients. Of those, 67,724 cases were witnessed by EMS (arrests after EMS arrival) and 3780 cases were unknown, both of which were excluded. Of the remaining 775,912 cases (281,601 bystander-witnessed cases and 494,311 non-witnessed cases), 13,194 were due to traffic collisions. Information on the first cardiac rhythm and bystander CPR was not obtained in 277 (2.10%) patients. After further excluding 10,830 patients who were not administered epinephrine, 2024 patients were considered eligible for our study.

### 3.2. Description of Baseline Features

The time of epinephrine administration was classified into the following quartiles: Q1 (6–21 min), Q2 (22–26 min), Q3 (27–34 min), and Q4 (35–60 min). Table 1 presents the patient and EMS characteristics of traumatic OHCA following traffic collision according to the timing of epinephrine administration, and Table 2 shows their outcomes. The median age of all patients was 67 (49–79) years, and the proportion of male individuals was 68.9%. The number of witnessed cases (345/466 [74.0%] in Q1, 356/472 [75.4%] in Q2, 401/577 [69.5%] in Q3, and 316/509 [62.1%] in Q4; *p* < 0.001) and the frequency of bystander CPR (175/466 [37.6%] in Q1, 152/472 [32.2%] in Q2, 186/577 [32.2%] in Q3, and 109/509 [21.4%] in Q4; *p* < 0.001) were both significantly different between the groups. The most common rhythms were the non-shockable rhythms (PEA and asystole), and the least common rhythms were VF/pVT (43/2024 [2.1%]). The percentage of patients with no advanced airway management was the highest in the Q1 group, and the rates of SGA were higher in the other three groups (*p* < 0.001).

### 3.3. Description of Outcomes

The one-month survival rates were 3.2% (15/466) in Q1, 1.1% (5/472) in Q2, 1.9% (11/577) in Q3, and 0.2% (1/509) in Q4. In multivariable logistic regression analysis, the probability of one-month survival significantly decreased in Q4 (adjusted OR, 0.07; 95% CI, 0.01–0.57) and as the time from the EMS call to epinephrine administration increased (*p*-value for trend = 0.009). Regarding the secondary outcomes, the adjusted ORs for pre-hospital ROSC were similar in Q2–Q4 (adjusted OR, 0.71 and 95% CI, 0.47–1.06 in Q2; adjusted OR, 0.89 and 95% CI, 0.61–1.28 in Q3; and adjusted OR, 0.66 and 95% CI, 0.43–1.00 in Q4). Unlike the other groups, Q1, the group that received epinephrine earliest (i.e., within 21 min of the call) included patients with favorable neurological outcomes (0.9% [4/466]). Logistic regression modeling with the timing of epinephrine administration as a continuous variable showed similar results (one-month survival: crude OR (95% CI), 0.93 (0.89–0.97); adjusted OR (95% CI), 0.93 (0.89–0.98); ROSC: crude OR (95% CI), 0.99 (0.97–1.00); adjusted OR (95% CI), 0.99 (0.97–1.00); one-month survival with favorable neurological outcomes: crude OR (95% CI), 0.77 (0.63–0.94); adjusted OR (95% CI), 0.70 (0.52–0.94)).

### 3.4. Factors Related to Mortality

The factors related to one-month survival after traumatic OHCA following traffic collision are shown in Table 3. Older patients aged 65–74 years (adjusted OR, 2.53; 95% CI, 1.09–5.91), male sex (adjusted OR, 3.98; 95% CI, 1.35–11.71), VF/pVT as first documented rhythm (adjusted OR, 8.18; 95% CI, 1.90–35.24), and PEA as first documented rhythm (adjusted OR, 2.99; 95% CI, 1.25–7.19) were factors associated with better outcomes. However, bystander CPR (adjusted OR, 0.74; 95% CI, 0.33–1.66) and ETI (adjusted OR, 2.01; 95% CI, 0.76–5.29) were not associated with better outcomes.

## 4. Discussion

Using data from a prospective OHCA registry, we demonstrated that delayed epinephrine administration was associated with decreased one-month survival after traumatic OHCA following traffic collisions. In addition, such patients with favorable one-month neurological outcomes had epinephrine administered only in the Q1 group.

The current evidence demonstrates that epinephrine improves the rates of ROSC but is not associated with improvements in neurologic or long-term outcomes [6]. Previous studies in the pre-hospital setting have shown that epinephrine only improved the rate of pre-hospital ROSC in traumatic OHCA cases following traffic collisions compared with no pre-hospital epinephrine administration [12]. In our study, after including patients who received pre-hospital epinephrine, pre-hospital ROSC did not change significantly at any time point. We also found that delayed epinephrine administration was associated with decreased one-month survival rates and no improvement in neurologic outcomes. Although ROSC may be achieved after organs are damaged by the ischemic effects of OHCA after delayed epinephrine administration, long-term or neurologic survival may not be possible due to severe organ damage. Epinephrine is thought to be beneficial to patients with cardiac arrest due to its ability to increase coronary perfusion pressure, potentially enhancing cardiac function. However, it may also reduce the cerebral flow and increase the myocardial oxygen demand [23,24]. The timing of drug delivery in relation to the onset of cardiac arrest may affect the benefit–harm balance of epinephrine.

Quick transport is beneficial for trauma patients. The EMS team should make rapid and accurate assessments to determine the required treatments and the appropriate hospital for each trauma patient, resulting in a shorter transport time [25,26] In the hospital setting, the resuscitation algorithm for traumatic OHCA was introduced systematically and immediately after aggressive treatment [27]. Definitive hemostatic treatment such as resuscitative thoracotomy and/or massive transfusion for hemorrhagic shock may be prioritized over epinephrine in hospital settings. Although guidelines remain unclear about epinephrine administration in patients with traumatic OHCA [28,29,30], epinephrine should be administered early in pre-hospital settings where early comprehensive intervention by EMS teams is limited.

Although it is easy to administer epinephrine early in hospital settings, only approximately 25% of the patients in this study received epinephrine within 21 min of OHCA occurrence. Even if earlier epinephrine administration improves survival from traumatic OHCA following a traffic collision, it may be difficult to administer epinephrine intravenously within 21 min at the scene of an accident. As ELSTs in Japan are allowed to administer epinephrine only intravenously, quicker routes for epinephrine administration such as intraosseous routes should be considered [31]. Generally, vascular access procedures and the administration of intravenous medications are advanced interventions that may lead to a longer time-to-epinephrine administration in certain EMS systems. The failure to secure intravenous cannulas can lead to reduced CPR quality or prolonged interruption of CPR. Incorporating intraosseous access and epinephrine administration into basic life support training may improve epinephrine administration in some EMS systems. Such advanced interventions, that only physicians can currently perform in Japan, may have a significant impact on the mortality of patients with traumatic OHCA.

According to a joint statement by the American College of Surgeons Committee on Trauma and the National Association of EMS Physicians [32], resuscitating patients with either penetrating or blunt trauma who are apneic, do not show signs of life, or have no detectable pulse is not recommended and should, instead, be pronounced dead at the accident location. Therefore, EMS teams also have the option of terminating resuscitation if they recognize OHCAs following traffic accidents in the field. However, in our multivariable analysis, it was revealed that PEA and VF/pVT were associated with improved survival, and resuscitation efforts for them should also be considered by the first documented rhythm. Further studies are needed to analyze the mortality in traumatic OHCA cases following traffic collisions with some of the above-mentioned criteria and better manage patients with serious injuries.

This study had some inherent limitations. First, the data on pre-event morbidities, in-hospital treatments such as trauma care (aortic cross-clamping, emergent thoracotomy, and fluid resuscitation) [27], hospital staffing, and post-cardiac arrest care [31] were missing from the registry. Second, our results may not be fully applicable in other countries, which have different EMS and medical systems wherein paramedics perform some advanced interventions. Therefore, further investigations of other cohorts are needed to address the generalizability and to confirm these associations. Third, there may have been unmeasured confounding factors that could have influenced the association between traumatic OHCA following traffic collision and the outcomes. Early epinephrine administration may also be a surrogate parameter for a team’s overall speed and experience [10]. In addition, we could not determine the detailed injured area or severity from the registry or whether the patients were drivers/passengers, motorcyclists, cyclists, or pedestrians. Finally, similar to all epidemiological studies, ascertainment bias, data integrity, and validity were potential limitations. The use of uniform data collection methods based on Utstein Style guidelines for reporting cardiac arrest, a population-based design, and a large sample size to gather all known OHCAs in Japan would minimize these potential sources of bias.

Although overall survival from traumatic OHCA was poor, small increases in survival rates may have a significant impact on public health. Providers should consider using methods for early epinephrine administration during the resuscitation of patients with OHCA following traffic collisions. However, they should prioritize the transportation to the hospital in case the administration is delayed, in order to be able to treat the definite injury. Further randomized control studies are needed to confirm these associations, considering the differences between pre-hospital and hospital settings. Furthermore, the importance of national preventive policies and individual responsibilities should be emphasized to avoid traffic collisions.

## 5. Conclusions

In this population, we found that delayed epinephrine administration was associated with decreased one-month survival after traumatic OHCA following traffic collisions. However, this study had some limitations, and the findings should be carefully interpreted and further investigated.

## Figures and Tables

**Figure 1 jcm-11-03564-f001:**
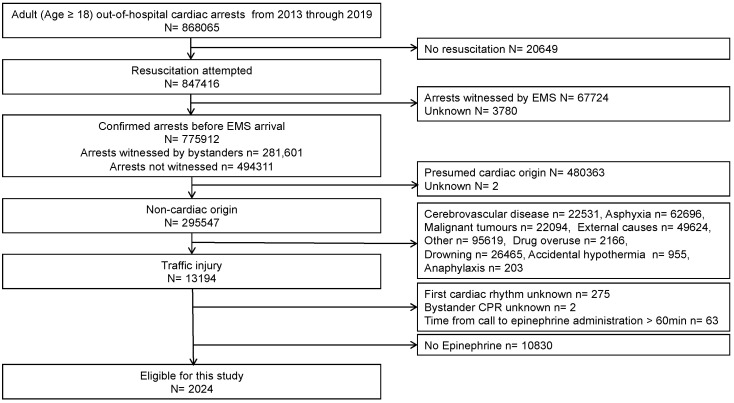
The patient selection flow chart. EMS, emergency medical services.

**Table 1 jcm-11-03564-t001:** The characteristics of patients with epinephrine administration.

			Epinephrine Administration (min)				
		Total	Q1 (6–21)	Q2 (22–26)	Q3 (27–34)	Q4 (35–60)	
		n = 2024	n = 466	n = 472	n = 577	n = 509	*p*-Value
Year of onset	2013	248 (12.3%)	60 (12.9%)	61 (12.9%)	65 (11.3%)	62 (12.2%)	0.026
	2014	264 (13.0%)	54 (11.6%)	67 (14.2%)	61 (10.6%)	82 (16.1%)	
	2015	237 (11.7%)	70 (15.0%)	56 (11.9%)	58 (10.1%)	53 (10.4%)	
	2016	295 (14.6%)	57 (12.2%)	54 (11.4%)	101 (17.5%)	83 (16.3%)	
	2017	315 (15.6%)	66 (14.2%)	68 (14.4%)	97 (16.8%)	84 (16.5%)	
	2018	316 (15.6%)	74 (15.9%)	74 (15.7%)	97 (16.8%)	71 (13.9%)	
	2019	349 (17.2%)	85 (18.2%)	92 (19.5%)	98 (17.0%)	74 (14.5%)	
Age, median (IQR), in years		67.00 (49.00–79.00)	70.00 (53.00–80.00)	70.00 (51.00–80.00)	66.00 (49.00–79.00)	61.00 (44.00–75.00)	<0.001
Age group, n (%), in years	18–64	906 (44.8%)	172 (36.9%)	190 (40.3%)	266 (46.1%)	278 (54.6%)	<0.001
	65–74	421 (20.8%)	113 (24.2%)	93 (19.7%)	119 (20.6%)	96 (18.9%)	
	>75	697 (34.4%)	181 (38.8%)	189 (40.0%)	192 (33.3%)	135 (26.5%)	
Sex (male), n (%)		1394 (68.9%)	300 (64.4%)	309 (65.5%)	412 (71.4%)	373 (73.3%)	0.004
Arrest witnessed by bystanders, n (%)		1418 (70.1%)	345 (74.0%)	356 (75.4%)	401 (69.5%)	316 (62.1%)	<0.001
First documented rhythm, n (%)	VF/pVT	43 (2.1%)	15 (3.2%)	10 (2.1%)	11 (1.9%)	7 (1.4%)	<0.001
	PEA	920 (45.5%)	278 (59.7%)	252 (53.4%)	252 (43.7%)	138 (27.1%)	
	Asystole	1061 (52.4%)	173 (37.1%)	210 (44.5%)	314 (54.4%)	364 (71.5%)	
Pre-hospital physician involvement		374 (18.5%)	72 (15.5%)	83 (17.6%)	104 (18.0%)	115 (22.6%)	0.031
Bystander CPR, n (%)		622 (30.7%)	175 (37.6%)	152 (32.2%)	186 (32.2%)	109 (21.4%)	<0.001
Advanced airway management, n (%)	ETI	184 (9.1%)	35 (7.5%)	25 (5.3%)	58 (10.1%)	66 (13.0%)	<0.001
	SGA	939 (46.4%)	200 (42.9%)	229 (48.5%)	267 (46.3%)	243 (47.7%)	
	None	901 (44.5%)	231 (49.6%)	218 (46.2%)	252 (43.7%)	200 (39.3%)	
Time from EMS call to contact with patient, min, median (IQR)		9.00 (7.00–13.00)	7.00 (6.00–9.00)	9.00 (7.00–11.00)	10.00 (8.00–13.00)	14.00 (10.00–18.00)	<0.001
Call to CPR, min, median (IQR)		11.00 (8.00–15.00)	8.00 (6.00–9.00)	9.00 (7.50–12.00)	12.00 (9.00–15.00)	18.00 (12.00–25.00)	<0.001

ETI, endotracheal intubation; SGA, supraglottic airway; VF, ventricular fibrillation; pVT, pulseless ventricular tachycardia; PEA, pulseless electrical activity; IQR, interquartile range; EMS, emergency medical services.

**Table 2 jcm-11-03564-t002:** The comparison of the primary and secondary outcomes of epinephrine administration.

	Epinephrine Administration (min)				
	Q1 (6–21)	Q2 (22–26)	Q3 (27–34)	Q4 (35–60)	*p* for Trend
	n = 466	n = 472	n = 577	n = 509	
* One-month survival, n (%)	15 (3.2%)	5 (1.1%)	11 (1.9%)	1 (0.2%)	
Crude OR (95% CI)	Reference	0.32 (0.12–0.89)	0.58 (0.27–1.28)	0.06 (0.01–0.45)	0.002
Adjusted OR (95% CI) *	Reference	0.38 (0.13–1.06)	0.64 (0.28–1.47)	0.07 (0.01–0.57)	0.009
Pre-hospital ROSC, n (%)	64 (13.7%)	47 (10.0%)	70 (12.1%)	47 (9.2%)	
Crude OR (95% CI)	Reference	0.69 (0.47–1.04)	0.87 (0.60–1.25)	0.64 (0.43–0.95)	0.083
Adjusted OR (95% CI) *	Reference	0.71 (0.47–1.06)	0.89 (0.61–1.28)	0.66 (0.43–1.00)	0.132
Favorable neurological outcomes, n (%)	4 (0.9%)	0 (0.0%)	0 (0.0%)	0 (0.0%)	
Crude OR (95% CI)	Reference	NA	NA	NA	NA
Adjusted OR (95% CI) *	Reference	NA	NA	NA	NA

OR, odds ratio; CI, confidence interval; NA, not applicable. * We performed the Hosmer–Lemeshow test and calculated the area under the receiver operator characteristic (ROC) curve to determine the model fit and discrimination for the primary outcome (goodness-of-fit, *p* = 0.7272, area under ROC curve = 0.8138, N = 2024).

**Table 3 jcm-11-03564-t003:** The factors associated with one-month survival.

		All (n)	One-Month Survival (n)	(%)	Crude OR	95% CI	Adjusted OR	95% CI
Year of onset		2024	32	1.58	0.98	(0.82–1.17)	0.96	(0.80–1.15)
Age group, n (%)	18–64 years	906	12	1.32	(ref)		(ref)	
	65–74 years	421	12	2.85	2.19	(0.97–4.91)	2.53	(1.09–5.91)
	≥75 years	697	8	1.15	0.87	(0.35–2.13)	1.01	(0.39–2.56)
Sex, n (%)	Female	630	4	0.63	(ref)		(ref)	
	Male	1394	28	2.01	3.21	(1.12–9.18)	3.98	(1.35–11.71)
Witness, n (%)	Arrests witnessed by bystanders	1418	23	1.62	1.09	(0.50–2.38)	1.13	(0.50–2.57)
	Arrests not witnessed	606	9	1.49	(ref)		(ref)	
First documented rhythm, n (%)	VF/pVT	43	3	6.98	11.29	(2.82–45.29)	8.18	(1.90–35.24)
	PEA	920	22	2.39	3.69	(1.57–8.68)	2.99	(1.25–7.19)
	Asystole	1061	7	0.66	(ref)		(ref)	
Bystander CPR, n (%)	No	1402	23	1.64	(ref)		(ref)	
	Yes	622	9	1.45	0.88	(0.40–1.91)	0.74	(0.33–1.66)
Pre-hospital physician involvement, n (%)	No	1650	25	1.52	(ref)		(ref)	
	Yes	374	7	1.87	1.24	(0.53–2.89)	1.12	(0.46–2.74)
Advanced airway management, n (%)	ETI	184	6	3.26	1.56	(0.62–3.97)	2.01	(0.76–5.29)
	SGA	939	7	0.75	0.35	(0.15–0.83)	0.37	(0.15–0.90)
	None	901	19	2.11	(ref)		(ref)	

VF, ventricular fibrillation; pVT, pulseless ventricular tachycardia; PEA, pulseless electrical activity; ETI, endotracheal intubation; SGA, supraglottic airway; CPR, cardiopulmonary resuscitation; EMS, emergency medical service; OR, odds ratio; CI, confidence interval; NA, not applicable; ref, reference.

## Data Availability

The data that support the findings of this study are available from the All-Japan Utstein Registry; restrictions apply to the availability of these data, which were used under license for the current study, and are therefore not publicly available. However, data are available from the authors upon reasonable request and with permission from the All-Japan Utstein Registry.

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
