# Peer review of "Association between Timing of Epinephrine Administration and Outcomes of Traumatic Out-of-Hospital Cardiac Arrest following Traffic Collisions"

_jcm, 2022, doi:10.3390/jcm11123564_

Round 1
Reviewer 1 Report
Brief summary
The aim of this study was to evaluate the effects of epinephrine administration timing on patients with out-of-hospital cardiac arrest (OHCA) following traffic collisions.
Findings
Delayed epinephrine administration was associated with decreased one-month survival after traumatic OHCA following traffic collisions. Additionally, only four patients (0.9%) who received the earliest epinephrine administration showed satisfactory neurological outcomes.
Strengths
Although overall survival from traumatic OHCA was poor, small increases in survival rates may have a significant impact on public health.
Minor issues
There are some typographic errors. There are very long sentences. To improve readability, consider breaking this into multiple sentences. The authors are encouraged to proof-read thoroughly the text before resubmission. English must be excellent.
Author Response
Response to Reviewer 1:
Brief summary
The aim of this study was to evaluate the effects of epinephrine administration timing on patients with out-of-hospital cardiac arrest (OHCA) following traffic collisions.
Findings
Delayed epinephrine administration was associated with decreased one-month survival after traumatic OHCA following traffic collisions. Additionally, only four patients (0.9%) who received the earliest epinephrine administration showed satisfactory neurological outcomes.
Strengths
Although overall survival from traumatic OHCA was poor, small increases in survival rates may have a significant impact on public health.
Minor issues
There are some typographic errors. There are very long sentences. To improve readability, consider breaking this into multiple sentences. The authors are encouraged to proof-read thoroughly the text before resubmission. English must be excellent.
Response: Thank you for this thorough review. Some redundant expressions have been removed. We have asked Editage for improving the readability of our article.
We have edited the 4th paragraph in the discussion (page 9) as follows:
“Although it is easy to administer epinephrine early in hospital settings, only approximately 25% of the patients in this study received epinephrine within 21 min of OHCA occurrence. Even if earlier epinephrine administration improves survival from traumatic OHCA following traffic collision, it may be difficult to administer epinephrine intravenously within 21 min at the scene of an accident. As ELSTs in Japan are only allowed to administer epinephrine onlyintravenously, quicker routes for epinephrine administration, such as intraosseous routes, should be considered [31]. The failure to perform advanced interventions can lead to reduced CPR quality or prolonged interruption of CPR. As establishing vascular access is essential for the administration of fluids or drugs, intraosseous access may replace intravenous access if establishing an intravenous route rapidly is difficult [28]. Generally, vascular access procedures and administration of intravenous medications are advanced interventions that may lead to a longer time-to-epinephrine administration in certain EMS systems, wherein procedures performed by paramedics at accident scenes are limited. The failure to secure intravenous cannulas can lead to reduced CPR quality or prolonged interruption of CPR. Incorporating intraosseous access and epinephrine administration into basic life support trainings may improve epinephrine administration in some EMS systems. Such advanced interventions, that only physicians can currently perform in Japan, may have a significant impact on the mortality of patients with traumatic OHCA. “

Reviewer 2 Report
Aim of this study was to evaluate the association between the timing of the first dose of epinephrine and outcomes of traumatic OHCA following traffic collisions, utilizing data from the All-Japan Utstein Registry, a prospective, population-based registry of OHCA.
Use of epinephrine in OHCA is still controversial, despite years of use and inclusion in resuscitation guidelines, in particular, epinephrine does not seem to improve neurologic outcomes. The data are mainly derived by multiple retrospective analyses and meta-analyses.
The aim of the present study represents an important question, the use of quality data from a prospective, large and well-organized registry (as described in paragraph 2.3) represents an important strength.
Statistical analysis is sound.
the paper is well written.
Comments:
- In this study, authors focused on a specific question about the use of epinephrine, i.e. “timing”. However, some general considerations should be included in the introduction and discussion, see Bornstein K, Long B, Porta AD, Weinberg G. After a century, Epinephrine's role in cardiac arrest resuscitation remains controversial. Am J Emerg Med. 2021 Jan;39:168-172. doi: 10.1016/j.ajem.2020.08.103. Epub 2020 Oct 21. PMID: 33162264.
- I could not find details about the dosage of administrated epinephrine.
- There is no information about the presence of pre-event morbidities, these might play an important role for survival and outcomes.
- As pointed out by authors, early administration of epinephrine depends upon the presence of ELSTs. The ELST per se might perform a better early support, thus it is impossible to associate a better survival rate only to the use of early epinephrine rather than a more specialized early comprehensive intervention. This point should be discussed.
- Authors suggest that a priority is to transfer the patient to the hospital in case the administration delayed, in order to be able to treat the definite injury. I agree. In this perspective, if I may, given the poor outcome in OHCA following traffic collisions, it should also be emphasized the importance of national preventive policies and individual sensibilities to avoid road accidents.
- given the comments above and the limitations described by the authors themselves, the conclusions should be more cautious.
Author Response
Response to Reviewer 2:
Comments and Suggestions for Authors
Aim of this study was to evaluate the association between the timing of the first dose of epinephrine and outcomes of traumatic OHCA following traffic collisions, utilizing data from the All-Japan Utstein Registry, a prospective, population-based registry of OHCA.
Use of epinephrine in OHCA is still controversial, despite years of use and inclusion in resuscitation guidelines, in particular, epinephrine does not seem to improve neurologic outcomes. The data are mainly derived by multiple retrospective analyses and meta-analyses.
The aim of the present study represents an important question, the use of quality data from a prospective, large and well-organized registry (as described in paragraph 2.3) represents an important strength.
Statistical analysis is sound. the paper is well written.
Response: Thank you for your thorough review. Our responses to your queries are as follows.
Comments:
- In this study, authors focused on a specific question about the use of epinephrine, i.e. “timing”. However, some general considerations should be included in the introduction and discussion, see Bornstein K, Long B, Porta AD, Weinberg G. After a century, Epinephrine's role in cardiac arrest resuscitation remains controversial. Am J Emerg Med. 2021 Jan;39:168- 172. doi: 10.1016/j.ajem.2020.08.103. Epub 2020 Oct 21. PMID: 33162264.
Response: Thank you. According to your suggestion, we have added the information to the introduction (lines 43–45): “Although epinephrine is one of the most widely used resuscitation drugs worldwide, the current literature suggests that epinephrine may provide benefit in certain situations [6]. Its benefits and risks remain controversial in the context of trauma [7,8].”
We have also made changes to the discussion (2nd paragraph):
Current evidence demonstrates epinephrine improves rates of ROSC but is not associated with improvements in neurologic or long-term outcomes [6]. Previous studies in the pre-hospital setting have shown that epinephrine only improved the rate of pre-hospital ROSC in traumatic OHCA cases following traffic collisions compared with no pre-hospital epinephrine administration [12]. In our study, after including patients who received pre-hospital epinephrine, pre-hospital ROSC did not change significantly at any time point. We also found that delayed epinephrine administration was associated with decreased one-month survival rates and no improvement in neurologic outcomes. Although ROSC may be achieved after organs are damaged by the ischemic effects of OHCA after delayed epinephrine administration, long-term or neurologic survival may not be possible due to severe organ damage. Epinephrine is thought to be beneficial to patients with cardiac arrest due to its ability to increase coronary perfusion pressure, potentially enhancing cardiac function. However, it may also reduce cerebral flow and increase myocardial oxygen demand [23,24]. The timing of drug delivery in relation to the onset of the cardiac arrest may affect the benefit–harm balance of epinephrine.
Reference
- Bornstein K, Long B, Porta AD, Weinberg G. After a century, Epinephrine's role in cardiac arrest resuscitation remains controversial. Am J Emerg Med. 2021 Jan;39:168- 172. doi: 10.1016/j.ajem.2020.08.103. Epub 2020 Oct 21. PMID: 33162264.
- I could not find details about the dosage of administrated epinephrine.
Response: We apologize for this omission. We have now included this in the methods section. (lines 91–92): “Patients aged ≥18 years received ≥1 dose of adrenaline (1 mg bolus) during resuscitation.”
- There is no information about the presence of pre-event morbidities, these might play an important role for survival and outcomes.
Response: Unfortunately, The All-Japan Utstein Registry date did not include such information, so we revised our limitations (lines 293–296).
“First, data on pre-event morbidities, in-hospital treatments, such as trauma care (aortic cross-clamping, emergent thoracotomy, and fluid resuscitation) [27], hospital staffing, and post-cardiac arrest care [31], were missing from the registry.”
- As pointed out by authors, early administration of epinephrine depends upon the presence of ELSTs. The ELST per se might perform a better early support, thus it is impossible to associate a better survival rate only to the use of early epinephrine rather than a more specialized early comprehensive intervention. This point should be discussed.
Response: Thank you. As you pointed out, ELSTs are considered to have more knowledge about trauma and resuscitation, see references 25 and 26, although they are allowed to perform some (limited) advanced pre-hospital interventions in Japan. We added the following sentences in the methods and discussion sections to emphasize these.
Methods (lines 87–89)
“In Japan, EMS personnel are not allowed to perform advanced interventions, such as drug administration (except for adrenaline) and needle decompression for tension pneumothorax.”
Discussion 3rd and 4th paragraphs:
“Quick transport is beneficial for trauma patients. The EMS team should make rapid and accurate assessments to determine required treatments and the appropriate hospital for each trauma patient, resulting in shorter transport time. [25, 26] In the hospital setting, the resuscitation algorithm for traumatic OHCA was introduced systematically and immediately after aggressive treatment [27]. Definitive hemostatic treatment, such as resuscitative thoracotomy and/or massive transfusion for hemorrhagic shock, may be prioritized over epinephrine in hospital settings. Although guidelines remain unclear about epinephrine administration in patients with traumatic OHCA [28–30], epinephrine should be administered early in pre-hospital settings where early comprehensive intervention by EMS teams is limited.
Although it is easy to administer epinephrine early in hospital settings, only approximately 25% of the patients in this study received epinephrine within 21 min of OHCA occurrence. Even if earlier epinephrine administration improves survival from traumatic OHCA following traffic collision, it may be difficult to administer epinephrine intravenously within 21 min at the scene of an accident. As ELSTs in Japan are only allowed to administer epinephrine onlyintravenously, quicker routes for epinephrine administration, such as intraosseous routes, should be considered [31]. The failure to perform advanced interventions can lead to reduced CPR quality or prolonged interruption of CPR. As establishing vascular access is essential for the administration of fluids or drugs, intraosseous access may replace intravenous access if establishing an intravenous route rapidly is difficult [28]. Generally, vascular access procedures and administration of intravenous medications are advanced interventions that may lead to a longer time-to-epinephrine administration in certain EMS systems, wherein procedures performed by paramedics at accident scenes are limited. The failure to secure intravenous cannulas can lead to reduced CPR quality or prolonged interruption of CPR. Incorporating intraosseous access and epinephrine administration into basic life support trainings may improve epinephrine administration in some EMS systems. Such advanced interventions, that only physicians can currently perform in Japan, may have a significant impact on the mortality of patients with traumatic OHCA.”
References
- Improved outcomes for out-of-hospital cardiac arrest patients treated by emergency life-saving technicians compared with basic emergency medical technicians: A JCS-ReSS study report. Naito H, Yumoto T, Yorifuji T, Tahara Y, Yonemoto N, Nonogi H, Nagao K, Ikeda T, Sato N, Tsutsui H.  2020 Aug;153:251-257. doi: 10.1016/j.resuscitation.2020.05.007
- Prehospital emergency life-saving technicians promote the survival of trauma patients: A retrospective cohort study. Nishimura T, Nojima T, Naito H, Ishihara S, Nakayama S, Nakao A. Am J Emerg Med. 2022 Jun;56:218-222. doi: 10.1016/j.ajem.2022.04.004
- Authors suggest that a priority is to transfer the patient to the hospital in case the administration delayed, in order to be able to treat the definite injury. I agree. In this perspective, if I may, given the poor outcome in OHCA following traffic collisions, it should also be emphasized the importance of national preventive policies and individual sensibilities to avoid road accidents.
Response: Thank you for your comments. According to your advice, we have revised the sentences as follows in the final paragraph of the discussion. “Although overall survival from traumatic OHCA was poor, small increases in survival rates may have a significant impact on public health. Providers should consider using methods for early epinephrine administration during resuscitation of patients with OHCA following traffic collisions. However, they should prioritize the transportation to the hospital in case the administration is delayed, in order to be able to treat the definite injury. Further randomized control studies are needed to confirm these associations, considering the differences between pre-hospital and hospital settings. Furthermore, the importance of national preventive policies and individual responsibilities should be emphasized to avoid traffic collisions.”
- given the comments above and the limitations described by the authors themselves, the conclusions should be more cautious.
Response: Thank you for pointing this out. We have revised our conclusion as follows (lines 320–323): “In this population, we found that delayed epinephrine administration was associated with decreased one-month survival after traumatic OHCA following traffic collisions. Additionally, only four patients (0.9%) who received the earliest epinephrine administration showed satisfactory neurological outcomes. However, this study had some limitations, and the findings should be carefully interpreted and further investigated.”

Reviewer 3 Report
I would like you congratulate the authors for their through and interesting work. In my understanding the general role on epinephrine in traumatic OHCA is not well established, but I think it is still interesting to study the timing of administration.
I have few comments I would like to address to the authors:
1. Patient characteristics and clinical scenario is quite different between the four quartiles. Age, witness arrest, first documented rhythm, bystander CRP etc.. for example median Call to CPR time was 8 min for Q1 and 18 min for Q4; this has great impact on survival.
2. How can you correct for many variables with such a low event rate (which is essentially 30-day survival)? I am not sure that the adjustment process is clear enough.
3. Table 3 in quite uncomfortable to read due to editing.
4. The authors concluded that “delayed epinephrine administration was associated with decrease one-month survival…”, I am not sure that such association can be made.
Author Response
Response to Reviewer 3:
Comments and Suggestions for Authors
I would like you congratulate the authors for their through and interesting work. In my understanding the general role on epinephrine in traumatic OHCA is not well established, but I think it is still interesting to study the timing of administration.
I have few comments I would like to address to the authors:
Response: Thank you for your thorough review. Our responses to your queries are as follows.
- Patient characteristics and clinical scenario is quite different between the four quartiles. Age, witness arrest, first documented rhythm, bystander CRP etc.. for example median Call to CPR time was 8 min for Q1 and 18 min for Q4; this has great impact on survival.
Response: As you noted, patient background was different between each group. Therefore, we used multivariable logistic regression analysis with such variables to adjust for this, as described in the Methods section (see 2.5. Statistical analysis). We did not use “median call to CPR” as a variable because of its collinearity with the variable “time from 119 call to epinephrine administration”.
- How can you correct for many variables with such a low event rate (which is essentially 30-day survival)? I am not sure that the adjustment process is clear enough.
Response: Thank you for your comment. Indeed, when the event rate is low in multivariable analysis, it is preferable to include fewer variables. We performed the Hosmer-Lemeshow test and calculated the area under the receiver operator characteristic (ROC) curve to determine the model fit and discrimination for the primary outcome (goodness-of-fit, p=0.7272, area under ROC curve=0.8138, N=2,024).
In addition, we confirmed that the OR in this multivariable analysis was calculable and convergent. Therefore, we believe that our data fit the model well. We added this goodness-of-fit test description as a footnote in Table 2:
“We performed the Hosmer-Lemeshow test and calculated the area under the receiver operator characteristic (ROC) curve to determine the model fit and discrimination for the primary outcome (goodness-of-fit, p=0.7272, area under ROC curve=0.8138, N=2,024).”
- Table 3 in quite uncomfortable to read due to editing.
Response: Following your advice, we have revised Table 3 to improve readability.
- The authors concluded that “delayed epinephrine administration was associated with decrease one-month survival…”, I am not sure that such association can be made.
Response: Considering that the p-value for trend=0.009 for one-month survival (Table 2), we believe the rate is statistically associated with the timing of epinephrine administration. However, we could not show the causality betweendelayed epinephrine administration and decreased one-month survival rate; therefore, further research is needed. We added the following in our conclusion: “However, this study had some limitations, and the findings should be carefully interpreted and further investigated.”

Round 2
Reviewer 2 Report
Thank you for the satisfactory replies.